# Synthesis and Characterization of Pt(II) and Pd(II) Complexes with Planar Aromatic Oximes

Mikala Meadows [1], Lei Yang [2], Cody Turner [1], Mikhail Berezin [3], Sergiy Tyukhtenko [4] and Nikolay Gerasimchuk [1,*]

1   Department of Chemistry and Biochemistry, Missouri State University, 901 South National Avenue, Temple Hall 456, Springfield, MO 65897, USA
2   Department of Chemistry, University of Central Arkansas, 201 Donaghey Ave, Manion 23B, Conway, AR 72034, USA
3   Department of Radiology, School of Medicine, Washington University, 510 S. Kingshighway Blvd, St. Louis, MO 63110, USA
4   Center for Drug Discovery, Northeastern University, 360 Huntington Avenue, Boston, MA 02115, USA
*   Correspondence: nngerasimchuk@missouristate.edu; Tel.: +1-(417)-836-5165

**Abstract:** A series of four Werner-type complexes of Pd(II) and Pt(II) with planar, isomeric conjugated aromatic naphtoquinone oximes were synthesized for the first time. These ligands were 1-oxime-2-naphtoquinone (**HL$^1$**) and 2-oxime-1-napthoquinone (**HL$^2$**). Compounds were characterized using thermal analysis, spectroscopic methods, and X-ray analysis. TG/DSC data were collected for pure starting organic ligands, their complexes, and indicated vigorous exothermic decomposition with at ~155 °C for starting HL and ~350 °C for transition metal complexes. Crystal structures for two Pt compounds with 2-oxime-1-quinone were determined and revealed the formation of the *cis*-geometry complexes and incorporation of molecules of stoichiometric solvents in the lattice: acetonitrile and nitrobenzene. Both solvents of crystallization displayed attractive interactions between their C-H groups and the oxygen atoms of the nitroso groups in complexes, leading to short distances in those fragments. Despite the presence of solvents of inclusion, the overall structure motifs in both compounds represent 1D columnar coordination polymer, in which the PtL$_2$ units are held together via metallophilic interactions, thereby forming 'Pt-wires'. The Hirshfield surface analysis was performed for both crystallographically characterized complexes. The results showed intermolecular π–π stacking and Pt–Pt interactions among the planar units of both complexes. In addition, the analysis also verified the presence of hydrogen bonding interactions between the platinum unit and solvent molecules. Solid bulk powdery samples of both **PtL$^1$$_2$** and **PtL$^2$$_2$** demonstrated pronounced photoluminescence in the near infrared region of spectrum at ~980 nm, being excited in the range of 750–800 nm. The NIR emission was observed only for Pt-complexes and not for pure starting organic ligands or Pd-complexes. Additionally, synthesized Pt-naphtoquinone oximes do not show luminescence in solutions, which suggests the importance of a 1D 'metal wire' structure for this process.

**Keywords:** naphthoquinone oximes; platinum (II); palladium (II); crystal structures; Hirshfield surface analysis; solid state UV–visible spectroscopy; NIR-emission; thermal analysis

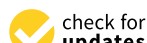



## 1. Introduction

Oximes represent versatile small molecules containing >C=N-OH fragments and are able to form stable coordination compounds (Werner-type) with a large variety of metal ions [1–4]. The structural chemistry of oximes and their metal complexes received great attention, owing to the applications of these compounds in analytical chemistry [5–8] as catalysts [9–11] and a wide spectrum of biological activity [12–14]. In the vast majority of these complexes, oximes act as deprotonated chelating ligands, since all of them represent weak organic acids, with the pKa ranging from ~3 in cyanoximes [15] to ~10 in aliphatic

oximes [16,17], covering a large swath of acid-base chemistry. There are several sub-classes of oximes based on the relationships between starting compounds and final products (Scheme 1). In all of them, compounds containing aromatic benzoid fragments, such as phenyl-, naphtyl-, and N-heteroaryl- moieties [1,18,19], were also made as variable R-groups. Among the oximes containing aromatic groups, isometic naphtoquinone oximes stemming from α-diketones take a special place (Scheme 2). These specific oximes were first used successfully by Ilinski and Knorre [20] as colorimetric and gravimetric organic reagents for the quantitative determination of transition metals cations [21–23]. Moreover, these two isomeric naphotquinone oximes have received considerable attention in the past, due to the postulated possibility that they can exist in two prototropic tautomeric *nitroso-/oxime* forms [24–29]. This likelihood was largely fueled by the inadequate and inconclusive use of available spectroscopic methods, such as IR and NMR spectroscopy in the early days. We were able to shed light on this issue with our recent decisive studies of these isomeric naphtoquinoneoximes, which we label herein as **HL$^1$** and **HL$^2$**, both in the solid states and solutions, and confirm that they, indeed, represent *oximes* [30]. However, being complexes with transition metals, these conjugated aromatic oximes become *nitroso*-anions, as evident from the significantly shorter N-O bonds, as compared to C-N bonds [31].

Despite being known and studied for a long time [20–23], the crystal structures of both organic ligands and some of their metal complexes were not known until recent years [30]. No chemical syntheses, properties, or characterization of two other representatives of the Ni-triade—Pd and Pt complexes—were reported. In this work, we present the structural and spectroscopic characterization of four complexes of Pd(II) and Pt(II) with two isomeric naphtoquinones.

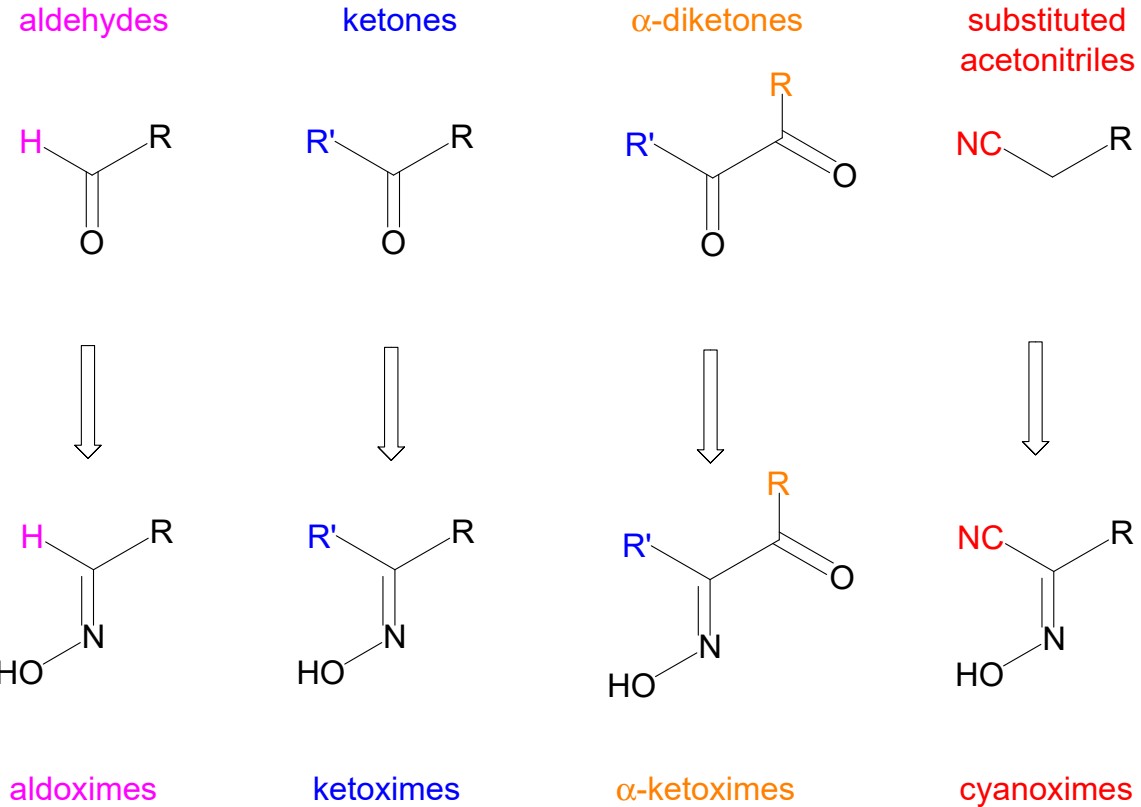

**Scheme 1.** Chemical structures of the C-oximes (where the oxime functionality = N-OH attached to the carbon atom) showing the relationship between the starting compounds and final oximes.

H(1NOH-2O) = **HL$^1$**      H(2NOH-1O) = **HL$^2$**

oximes

deprotonation during
complex formation

nitroso-compounds

1NO-2O = **L$^1$**          2NOH-1O = **L$^2$**

**Scheme 2.** Chemical structures of planar aromatic oximes and derived from them anions used in this study with their abbreviations.

## 2. The Experimental Part

### 2.1. General Considerations

The transition metals sources for the preparation of complexes were potassium tetrachloride salts $K_2PdCl_4$ and $K_2PtCl_4$, obtained from Pressure Chemicals. Pure organic ligands—isomeric naphthoquinone oximes—were obtained from Aldrich and Alfa Aesar and used as received. The organic solvents were all of ACS grade and obtained from commercial vendors and were used without additional purification. Elemental analyses determination was carried out by the combustion method on C, H, and N content at Atlantic Microlab.

### 2.2. Syntheses of Coordination Compounds

Both Pd(II) and Pt(II) complexes were obtained according to Scheme 3 using a two-step procedure. First, 0.250 g of organic ligand was dissolved in acetonitrile at room temperature in a 50 mL beaker, and an equivalent amount of aqueous solution (5 mL) of potassium carbonate was added, dropwise, under stirring. The colour of the reaction mixture changed to orange-brown, and gas evolved. At the second stage, 5 mL of stoichiometric amounts 0.72 mM of transition metals tetrachlorides as potassium salts (0.299 g for Pt and 0.236 g for Pd) were added dropwise under stirring at room temperature. In the case of Pd(II) complexes, the precipitation was complete within ~10 min, while for the Pt(II) compounds, the reaction mixtures were left in a dark for an overnight. In both cases, intensely colored fine precipitates were formed (Supporting Information S1,S2) and carefully gravity-filtered using paper filters. Then, precipitates were washed twice with 10 mL of distilled water and left to air dry in the dark. Yields of compounds varied, between 64–70% for Pt complexes and 86–92% for Pd complexes. It is important to note that the

recovery of these complexes took from 4 to 9 days because they form a very fine amorphous precipitates that require multiple sequential filtrations of mother liquors. Analytical data obtained in this way suggested a desirable 1:2 composition of bulk powders, with respect to metal-to-ligand ratio (Supporting Information S3–S5). Palladium(II) complexes, however, for no apparent reason, contained few occluded water molecules in their composition (Supporting Information S4,S5). It is interesting to observe in this case similarity with some Pd-cyanoximates that also contained solvent molecules of preparation/crystallization [32].

$$2\ HL\ +\ K_2CO_3\ \xrightarrow{\text{R.T.}}\ 2\ KL\ +\ CO_2\uparrow\ +\ H_2O$$

$$\text{in situ} \quad \Big\downarrow \quad +\ K_2MCl_4\ (M = Pd,\ Pt)$$

$$ML_2\downarrow\ +\ 4\ KCl$$

L =    1-oxime-2-quinone    2-oxime-1-quinone

**Scheme 3.** Two-step preparation of four Pd, Pt complexes of planar aromatic isometic naphto-quinone oximes.

*2.3. Spectroscopy*

The IR-spectra of all the compounds used in this work were recorded on a Bruker Vertex 70 FT-IR spectrophotometer as KBr pellets, or small amounts of powdered compounds were gently spread over Ge crystal using a Bruker Alfa FT-ATR instrument. The UV–visible spectra of initial ligands in solutions in organic solvents were recorded using a diode array HP 8354 spectrophotometer, operating in a 200–1100 nm range, in quartz cuvettes at 295 K. Diffusion reflectance spectra (SDR) for powders of four Pd, Pt complexes were recorded at room temperature with the help of a CARY 100 Bio spectrophotometer equipped with an integrating sphere (Labsphere). Photoluminescence spectra of powders of Pt-complexes in the range of 800–1500 nm were recorded on Horiba Quantum Master QM-8075-21C spectrophotometer equipped with InGaAs NIR detector.

*2.4. Thermal Analysis*

The TG/DSC traces for both ligands and four synthesized Pd, Pt complexes were recorded on the TA Instrument Q-600 (Delaware, USA) in an aluminum oxide crucible under pure argon flow, in the range between 30–900 °C at $100 \pm 1$ mL/min, and a heating rate of 10 °C/min. The crucible was calcined prior to each experiment using the propane torch. Data of the TG analyses were processed using the TA Universal Analysis software package.

### 2.5. Crystallographic Work

The MoK$\alpha$ ($\lambda$ = 0.71073 Å) radiation source was used in the structural characterization of synthesized compounds using a Bruker APEX2 diffractometer with 0.5° step and four omega runs of 364 frames each, covering a full sphere of reflections.

Growing crystals for all compounds reported in this work proved to be a challenging procedure, which led to very thin needles and plates of complexes (Supporting Information S6). The grown crystals of two solvated **PtL$^2_2$** complexes demonstrated a strong propensity for twinning and actually represented cracked multidomain species with a clearly dominant major component in more than 60% of reflections, followed by several smaller, and randomly oriented, domains.

Several crystalline specimens of **PtL$^2_2$** (acetonitrile solvate) were tried. The most suitable and less complex crystal (mozaicity 0.68°) was selected for data collection. Still, it turned out to be a six-component specimen, and only the first four were used for a successful structure solution and refinement. Thus, using the CELL_NOW _T program (from the Bruker suite), 2992 strong reflections were selected, with I > 20$\sigma$(I) from 1454 frames. Among these, 2581 reflections were allocated to domain #1, 1942 to domain #2, 361 reflections to domain #3, and 408 reflections to domain #4, with twin laws relating all components to the main one presented in Supporting Information S7. To determine the crystal structure, 1456 frames were collected (the total exposure time was 24.27 h). The frames were integrated with the Bruker SAINT software package, using a narrow-frame algorithm. The integration of the data using a triclinic unit cell yielded a total of 3234 reflections to a maximum $\theta$ angle of 25.00° (0.84 Å resolution), of which 3234 were independent (average redundancy 1.000, completeness = 92.3%, Rsig = 25.19%), and 1883 (58.23%) were greater than 2$\sigma$(F$^2$); Rint = 5.52%. The observed values of T$_{min}$ and T$_{max}$ transmission coefficients were 0.4602 and 0.7554, respectively. Structure was solved using direct methods. In the final refinement cycles, flat shapes of thermal ellipsoids for some atoms were mitigated by applying DELU (at 0.005 level) and SIMU (at 0.004 level) restraints.

Despite numerous trials, the best crystal of **PtL$^2_2$** (nitrobenzene solvate) selected for studies still turned out to be a multi-component specimen, which can be described as cracked twin (mozaicity 0.81°). Thus, using the CELL_NOW_T program from the Bruker software suite, we performed de-twinning of reflections in this dataset similarly to the previous case of the acetonitrile solvate and results are shown in Supporting Information S8. Twin laws relating all components to the main one presented herein, as well. Structure was solved using direct methods. In both reported structures herein, the data were corrected for absorption effects using the multi-scan method (SADABS) embedded into the TWINABS program of the instrument software suite [33]. Yet, despite all the efforts for accounting and indexing all the available reflections in de-twinned datasets, there were still a considerable number of Q-peaks located near the metal centers (Supporting Information S9,S10). Those 'ripples' of residual electron density were located at 1–1.5 Å distances from the Pt atoms and did not possess any chemical sense, but generated A- and B-type alerts in the final check CIF reports. The detected positive electron density on metal centers is indirect evidence for the partial formation of Pt(IV) oxidation states. The crystal and refinement data are summarized in Table 1, while the respective ASU content drawings were performed using the ORTEP3.v2 software [34]. Mercury [35] packages are shown in Figures 1–4. Determined crystal structures were submitted to the CCDC and received registration numbers 2,240,578 (acetonitrile solvate) and 2,240,577 (nitrobenzene solvate). Check CIF reports are presented at the end of the Supporting Information section in Supporting Information S38. Broad powder patterns of all four synthesized complexes were not informative, since the bulk samples represented amorphous fine precipitates.

**Table 1.** Crystal and refinement data for two Pt(II) naphtoquinone-oximate solvates.

| Parameter | Compound | |
|---|---|---|
| | $PtL^2_2 \cdot CH_3CN$ | $PtL^2_2 \cdot C_6H_5NO_2$ |
| Empirical formula | $C_{22}H_{15}N_3O_4Pt$ | $C_{26}H_{17}N_3O_6Pt$ |
| Formula weight, g/M | 580.46 | 662.52 |
| Temperature, K | 120(2) | 100(2) |
| Crystal system | triclinic | triclinic |
| Color/habitus | metallic black needle | green-black plate |
| Crystal size (mm) | $0.04 \times 0.05 \times 0.302$ | $0.042 \times 0.226 \times 0.408$ |
| Space group | P-1, #2 | P-1, #2 |
| Cell constants, Å/° | | |
| a | 6.679(8) | 6.707(2) |
| b | 10.897(13) | 12.203(4) |
| c | 13.866(17) | 14.016(5) |
| $\alpha$ | 97.279(14) | 92.739(5) |
| $\beta$ | 93.082(16) | 103.808(5) |
| $\gamma$ | 92.877(14) | 96.875(5) |
| Volume ($Å^3$) | 998(2) | 1102.5(6) |
| Z | 2 | 2 |
| $\rho_{calc}$ (g/cm$^3$) | 1.932 | 1.996 |
| $\mu$ (mm$^{-1}$) | 7.065 | 6.413 |
| F(000) | 556 | 640 |
| 2$\Theta$ range for data (°) | 2.96 to 50.00 | 3.38 to 53.1 |
| Index ranges: | $-8 < h < 8$ | $-8 < h < 8$ |
| | $-13 < k < 13$ | $-15 < k < 15$ |
| | $0 < l < 17$ | $-17 < l < 17$ |
| Reflections collected | 3234 | 13115 |
| Independent | 3234 | 4553 [R(int) = 0.079] |
| Data/restrains/parameters | 3234/48/272 | 4553 / 0 / 323 |
| Goodness-of-fit on $F^2$ | 0.762 | 0.973 |
| Final R indices: | 1883 [I>2σ(I)]; R1 = 0.0590 | 4553 [I>2σ(I)]; R1 = 0.0572 |
| | wR2 = 0.1153 | wR2 = 0.1430 |
| All data: | R1 = 0.1030; wR2 = 0.1264 | R1 = 0.0683; wR2 = 0.1515 |
| Peak/hole difference, e(A$^{-3}$) | 2.279 and $-1.68$ | 5.340 and $-3.981$ |
| Volume taken, A$^3$ (%) | 638.5 (63.9) | 742.53 (67.9) |

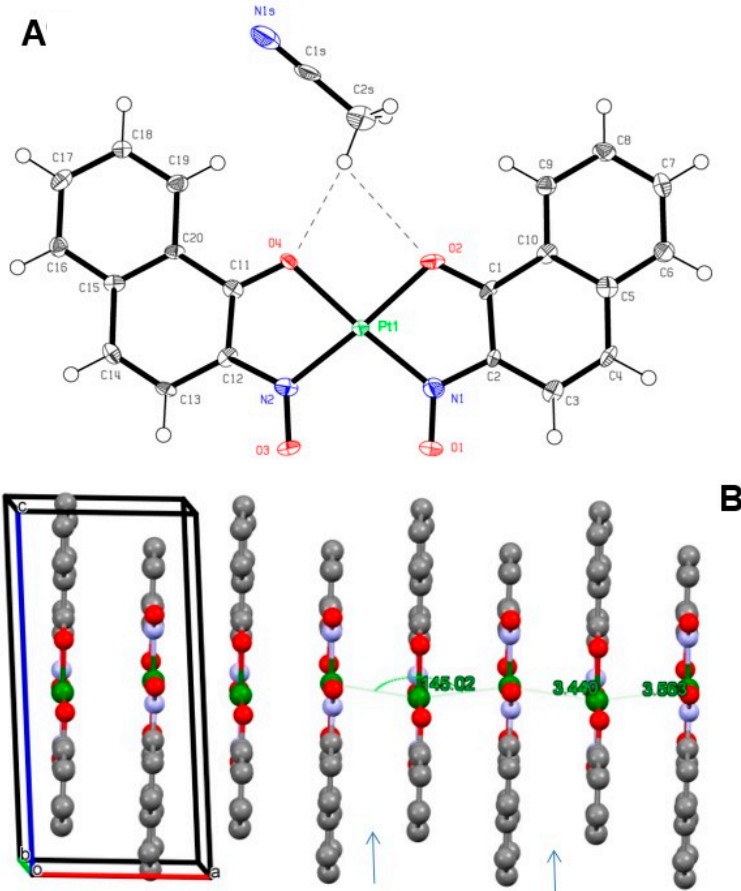

**Figure 1.** (**A**)—The ASU in the structure of $PtL_2^2 \times CH_3CN$ showing the numbering scheme in this stoichiometric solvate and its unusual orientation towards main residue with short C-H—O contacts; (**B**)–prospective view along *b*-direction of three-unit cells that were extended along *a*-direction to show slightly eclipsed π–π stacked head-to-tail dimers, indicated by arrows. Solvent molecules and H-atoms are omitted for clarity.

### 3. Results and Discussion

#### 3.1. Thermal Analysis

To our great surprise, no studies of the thermal behavior of isomeric naphtoquinone-oximes had been carried out before. In order to fill that gap, we recorded TG/DSC traces of **HL¹** and **HL²** in the range of 30 °C to 800 °C under a flow of high purity $N_2$ gas. First, we were not able to corroborate the phase transition temperatures for pure organic ligands reported earlier: no melting was detected. Instead, a rather sharp and fast decomposition in the range of 155–170 °C was observed within a narrow time frame (Table 2). The first step of decomposition of the initial naphtoquinone-oximes is the loss of water (Supporting Information S11,S13). Thus, the calculated/(found) amounts of one molecule of water are 10.79% (10.13%) for **HL¹** and 10.79% (12.42%) for **HL²**. The process is presumably condensation (or esterification?) of naphtoquinone-oximes to dimers. **HL¹** was found to be the most energetic compound among those studied, with releases over 190 kJ/mol, but within ~9.3 min. The final product of the thermal decomposition of pure organic ligands was glassy carbon, which formed sheets inside crucibles (Supporting Information S12,S14).

Both Pd and Pt metal complexes also demonstrate similar sharp and fast thermal decomposition, but at temperatures which were almost twice as high! This observation suggests that complexation with transition metals considerably improves the thermal stability of coordinated isomeric naphtoquinones (Table 2). Additionally, a sharp exothermic decomposition peak was observed, which evidenced a dramatic and quick energy release in an **L²**-based system in a very short time interval. This can be used for the potential

application of both **PdL$^2_2$** and **PtL$^2_2$** as heat activated high-energy compounds. Quite similar unexpected thermal behavior for several Cu(II) [36] and Pb(II) cyanoximates [37] had been observed earlier. The final products of the thermal decomposition of all studied metal complexes were spongy Pd and Pt metals (Supporting Information S15–S19).

**Table 2.** Results of thermal analysis studies of pure naphtoquinone-oximes and Pd, Pt-complexes, showing the parameters of their exothermic decomposition.

| Compound | F.W. | Exo-Peak (T °C) | E, Released (kJ/mol) | Time Interval (min) | Heat Release (J/sec) |
|---|---|---|---|---|---|
| **HL$^1$** | 173.2 | 156.7 | 192.1 | ~9.3 | 344 |
| **HL$^2$** | 173.2 | 168.9 | 98.9 | ~2 | 825 |
| **PtL$^2_2$** | 539.4 | 322.8 | 112.3 | ~3.8 | 492 |
| **PdL$_1$$^2$** | 450.7 | 304.3 | 119.6 | ~3 | 665 |
| **PdL$^2_2$** | 450.7 | 309.3 | 140.5 | ~2 | 2342 |

*3.2. Crystal Structures*

Both of the single crystals of Pt(II) complexes studied in this report contained stoichiometric molecules of solvents of crystallization: acetonitrile and nitrobenzene. These were found to be the most suitable solvents for crystal growth using the vapor diffusion method. Unfortunately, despite numerous attempts, we could not grow single crystals of Pd(II) and Pt(II) containing the aromatic napthoquinone **L$^1$** suitable for X-ray analysis (Scheme 2). The molecular structure and numbering scheme for the acetonitrile solvate **PtL$^2_2$·CH$_3$CN** is shown in Figure 1A, while the most important bond lengths and angles at the metal center in this structure are presented in Supporting Information S20. This complex represents a 1D polymer, in which individual units are held together by weak Pt–Pt metallophylic interactions and slightly slipped π–π staking interactions, as shown in Figure 1B, Supporting Information S21. A polymeric motif is formed by the dimers, with somewhat different inner/outer dimeric Pt–Pt distances of 3.440 and 3.563 Å, respectively. Inside the dimer, there is a "head-to-tail" orientation of **PtL$^2_2$** units. An angle between the Pt atoms in the formed 'metallic wire' is ~145° (Figure 1B: Supporting Information S22). The geometry of the aromatic oxime **L$^2$** in this Pt complex is presented in Table 3 and compared appropriately with pure oxime in its protonated state [30]. The solvent molecule of CH$_3$CN is not intercalating dimers, but rather connected to one **PtL$^2_2$** unit, which is contrary to the behavior of the nitrobenzene molecule in the structure described next. Acetonitrile forms short electrostatic contacts with oxygen atoms O2 and O4 of the naphthol part of the ligand with geometrical arrangements displayed in Supporting Information S23.

The molecular structure and numbering scheme for the nitrobenzene solvate **PtL$^2_2$·C$_6$H$_5$NO$_2$** is shown in Figure 2A, with bonds and angles involving the metal center displayed in Supporting Information S24. This complex also represents a 1D polymer with weak Pt–Pt metallophylic interactions and slipped π–π staking interactions, as shown in Figure 2B and Supporting Information S25. A polymeric motif is formed by dimers with marginally different inner/outer dimeric Pt–Pt distances of 3.301 and 3.536 Å, respectively. In the dimer, individual molecules **PtL$^2_2$** are positioned in a "head-to-tail" fashion (Supporting Information S25). An angle between the Pt atoms in the formed 'metallic wire' extended along the *a* direction is ~156° (Figure 2B: Supporting Information S26). The geometry of the aromatic oxime **L$^2$** in this Pt complex is also presented in Table 3. It should be noted that the coordination of the anionic ligand to Pt-centers in both complexes facilitated the adoption of the *nitroso*-form, compared to protonated **HL$^2$**, which is an *oxime* both in the solid state and solutions [30], as depicted in Scheme 2. Thus, the N-O bonds in both **[PtL$^2_2$]** (solv) are significantly shorter than C-N in the >C-N-O fragment (Table 3; Scheme 2). This situation is similar to Co(III) complexes with these ligands observed earlier [31]. The solvent molecule of nitrobenzene is actually intercalated between two dimers with two metallophilic Pt–Pt contacts (Figure 2;

Supporting Information S27). This solvent forms two sets of rather short, electrostatic in nature, C-H—O contacts with all four oxygen atoms of the ligand in the range of 2.63–2.69 Å and a wide range of contacts angles from 106° to 151° (Supporting Information S28).

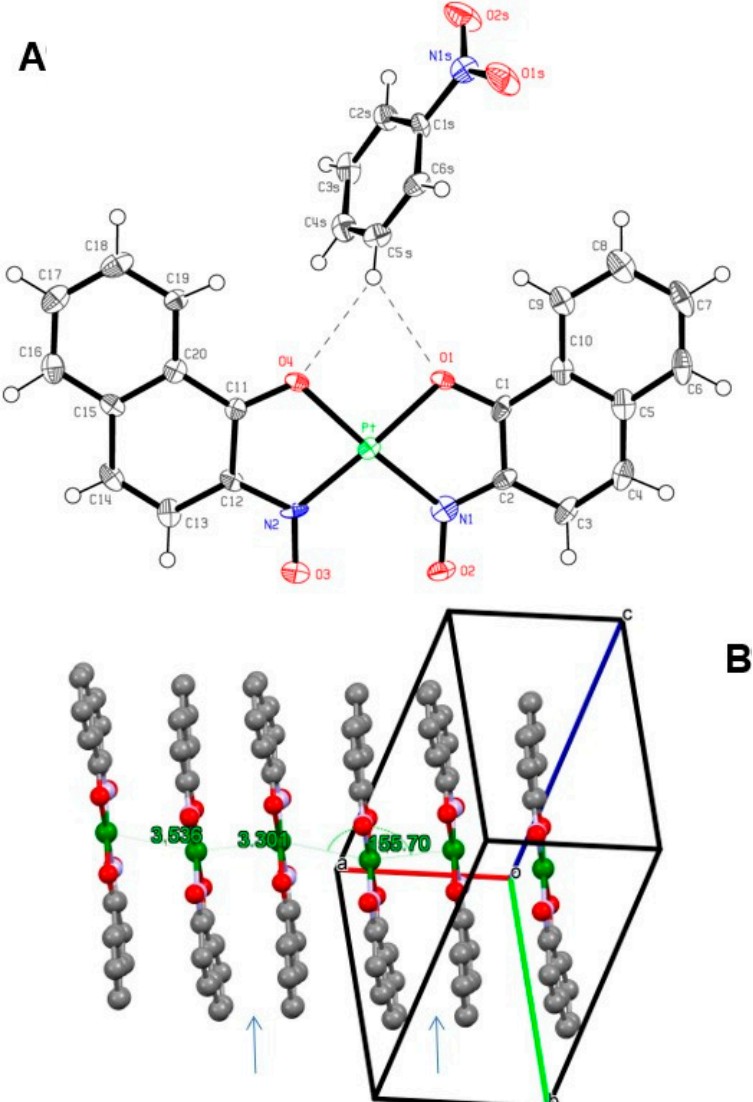

**Figure 2.** (**A**)—The ASU in the structure of $PtL_2^2 \times C_6H_5NO_2$ showing the numbering scheme in this solvate and its different orientation towards complex main residue with short C-H–O contacts; (**B**)—prospective view along unit cell diagonal [111] of three-unit cells that were extended along *a*-direction to show slightly eclipsed π–π stacked head-to-tail dimers indicated by arrows. Solvent molecules and H-atoms are omitted for clarity.

Central atoms in both **[PtL$^2_2$]** (solv) complexes adopt a planar trapezoid environment, with the Pt-N distances shorter and the Pt-O bonds and angles <N-Pt-N being considerably greater than 90°, with <O-Pt-O slightly above 80° (Supporting Information S20,S24). The chelate 'biting' angles <N1-Pt1-O2 and <N2-Pt1-O4 are 82.01° and 80.79° in the structure of **PtL$^2_2$·CH$_3$CN**, and same angles are 82.61° and 81.40° for **PtL$^2_2$, C$_6$H$_5$NO$_2$**. All the bond lengths and angles in the solvent molecules inside these complexes were normal.

The two anions in both structures adopt a *cis* orientation, relative to Pt-centers (Figures 1A and 2A). This is consistent with several similar geometries we have recently observed in Pt(II) complexes with cyanoximes [38–40]. This is different from numerous *trans* complexes of cyanoximes with light transition metals of 3d series: Ni$^{2+}$ [41], Cu$^{2+}$ [42,43],

and $Fe^{2+}$ [44,45]. The origin of such geometrical preferences for Pt complexes is not clear at the moment.

One of the most interesting findings in both structures was the presence of stoichiometric solvent molecules of solvents close to the nitroso- and/or naphtol oxygen atoms (Supporting Information S23,S27). This is an interesting observation, considering the other Pt-oximates in structures where the same feature was also found (Table 4). It appears that there are pronounced C-H—O contacts, most likely of electrostatic origin, at distances shorter than the sum of van der Waals radii of the involved atoms. In order to investigate this phenomenon further, we performed the Hirshfield surface analysis for both the **$PtL^2_2 \cdot CH_3CN$** and **$PtL^2_2 \cdot C_6H_5NO_2$** complexes reported herein. The aim was to examine the nature of the interactions and forces that necessitate such crystal packing.

**Table 3.** Selected bond lengths and angles in the structures of protonated **$HL^2$** and studied Pt solvates.

| Compound | Bonds, Å | Angles, ° |
|---|---|---|
| **$HL^2$ *** | C1-N1 = 1.304 | N1-C1-C2 = 113.4 |
| | 1.370 | C1-N1-O1 = 112.8 |
| | 1.227 | N1-O2-H(O) = 109.6 |
| | 1.481 | O2-C2-C1 = 121.1 |
| | 1.454 | O2-C2-C3 = 121.9 |
| | 1.339 | N1-C1-C10 = 126.0 |
| **$PtL^2_2 \cdot CH_3CN$** | C1-O1 = 1.326(11) | O2-C1-C2 = 122.0(13) |
| | C11-O4 = 1.302(15) | O2-C1-C10 = 119.6(13) |
| | C1-C2 = 1.45(2) | O1-N1-C2 = 117.8(11) |
| | C2-C3 = 1.428(19) | C3-C2-N1 = 124.6(12) |
| | C3-C4 = 1.397(19) | N1-C2-C1 = 111.4(12) |
| | C2-N1 = 1.431(17) | O4-C11-C12 = 119.8(13) |
| | C11-O4 = 1.302(15) | O4-C11-C20 = 122.8(12) |
| | C11-C12 = 1.450(19) | N2-C12-C11 = 112.4(12) |
| | C12-N2 = 1.383(17) | N2-C12-C13 = 124.8(12) |
| | C12-C13 = 1.474(19) | O3-N2-C12 = 119.8(11) |
| | C13-C14 = 1.386(19) | |
| | N1-O1 = 1.274(14) | |
| | N2-O3 = 1.301(14) | |
| **$PtL^2_2 \cdot C_6H_5NO_2$** | C1-O2 = 1.326(11) | O1-C1-C2 = 118.3(9) |
| | C11-O4 = 1.338(11) | O1-C1-C10 = 121.9(9) |
| | N1-O2 = 1.249(11) | O2-N1-C2 = 119.3(8) |
| | N2-O3 = 1.242(10) | N1-C2-C3 = 126.0(9) |
| | C1-C2 = 1.430(14) | N1-C2-C1 = 113.8(8) |
| | C11-C12 = 1.436(13) | O4-C11-C12 = 117.4(8) |
| | C2-N1 = 1.357(13) | O4-C11-C20 = 123.2(8) |
| | C12-N2 = 1.391(12) | N2-C12-C11 = 112.6(8) |
| | C2-C3 = 1.427(13) | N2-C12-C13 = 126.7(9) |
| | C3-C4 = 1.346(15) | O3-N2-C12 = 117.0(8) |
| | C11-C12 = 1.436(13) | |
| | C12-C13 = 1.411(13) | |

*—Data for the protonated, uncoordinated ligand taken from reference source [30].

**Table 4.** Some structural data for Pt-based oximates with co-crystallized solvents.

| Compound | Structure | Solvent | Geometrical Parameters, Å | | Refs. |
|---|---|---|---|---|---|
| | | | Pt—Pt | N-O—H-C(solv) | |
| [a] $[Pt(MCO)_2]_2$ | dimer | DMSO | 3.133 | 2.526, 2.535 | [37] |
| [b] $[Pt(PyrCO)_2]_2$ | dimer | DMSO | 3.207 | 2.737, 2.743 | [38] |
| $[PtL^2_2]$ | 1D polymer | $CH_3CN$ | 3.440, 3.563 | 2.543, 2.857 | this work |
| $[PtL^2_2]$ | 1D polymer | $C_6H_5NO_2$ | 3.301, 3.536 | 3.845, 2.994 | this work |

[a]—MCO stands for 2-oximino-2-cyano-N-morpholyl-acetamide [32].　[b]—PyrCO is 2-oximino-2-cyano-N-pyrrolidine-acetamide [39].

### 3.3. Spectroscopic Studies

3.3.1. Vibrational Spectra

For the identification of vibrations with participation of the >C-N-O fragment, we obtained isotopically labeled $^{15}N$ (at ~50% enrichment) starting naphtoquinone-oxime **HL$^1$** and its Pt, Pd complexes of **PdL$^1_2$** and **PtL$^1_2$**. We clearly observed splitting of $\nu$(C=N) and $\nu$(N-O) stretching bands and a decrease in their intensity provided their unambiguous assignment (Table 5; Figure 3).

**Table 5.** Vibrational frequencies (cm$^{-1}$) in the IR spectra of the starting aromatic ligands * and their palladium and platinum complexes. Assignment of the $\nu$ (N-O) and $\nu$ (C=N) bands are based on spectra of $^{15}N$ labeled samples (in brackets).

| Tentative Band Assignment | | | | |
|---|---|---|---|---|
| Complex | $\nu$ (C-H, aryl) | $\nu$ (C=C) | $\nu$ (C=N) | $\nu$ (N-O) |
| HL$^1$ | 3065, 3020 | 1617, 1605 | 1520 (1505) | 1067 (1057) |
| HL$^2$ | 3085, 2973 | 1622 | 1548 | 1062 |
| PdL$^1_2$ | 3047, 2918 | 1607, 1589 | 1394 | 1088 |
| PdL$^2_2$ | 3058 | 1603, 1587 | 1396 | 1089 |
| PtL$^2_2$ | 3065 | 1606, 1587 | 1406 (1395) | 1100 |
| PtL$^1_2$ | 3058 | 1611 | 1402 (1392) | 1093 (1079) |

*—broad $\nu$(O-H) vibrations in spectra of uncomplexed ligands are observed above 3200 cm$^{-1}$ and not listed.

The introduction of isotopic labeling of oximes/nitroso compounds was pioneered by H. Köhler [46,47], back in a day when X-ray crystallography was not readily accessible tool and coordination compounds were difficult to crystallize. The Na$^{15}NO_2$ can be conveniently used for labeling by means of the nitrosation reaction at ~0 °C under acidic conditions [48,49]. Thus, in many cases, deduced structural considerations were based on the extensive use of vibrational spectroscopy, which allowed for the determination of the binding mode of ligands using correct assignment of vibrations of the >C-N-O fragment. We successfully used such an approach in the past, identifying the vibrations of the oxime/nitroso groups in the IR-spectra of ligands and their metal complexes [48,49].

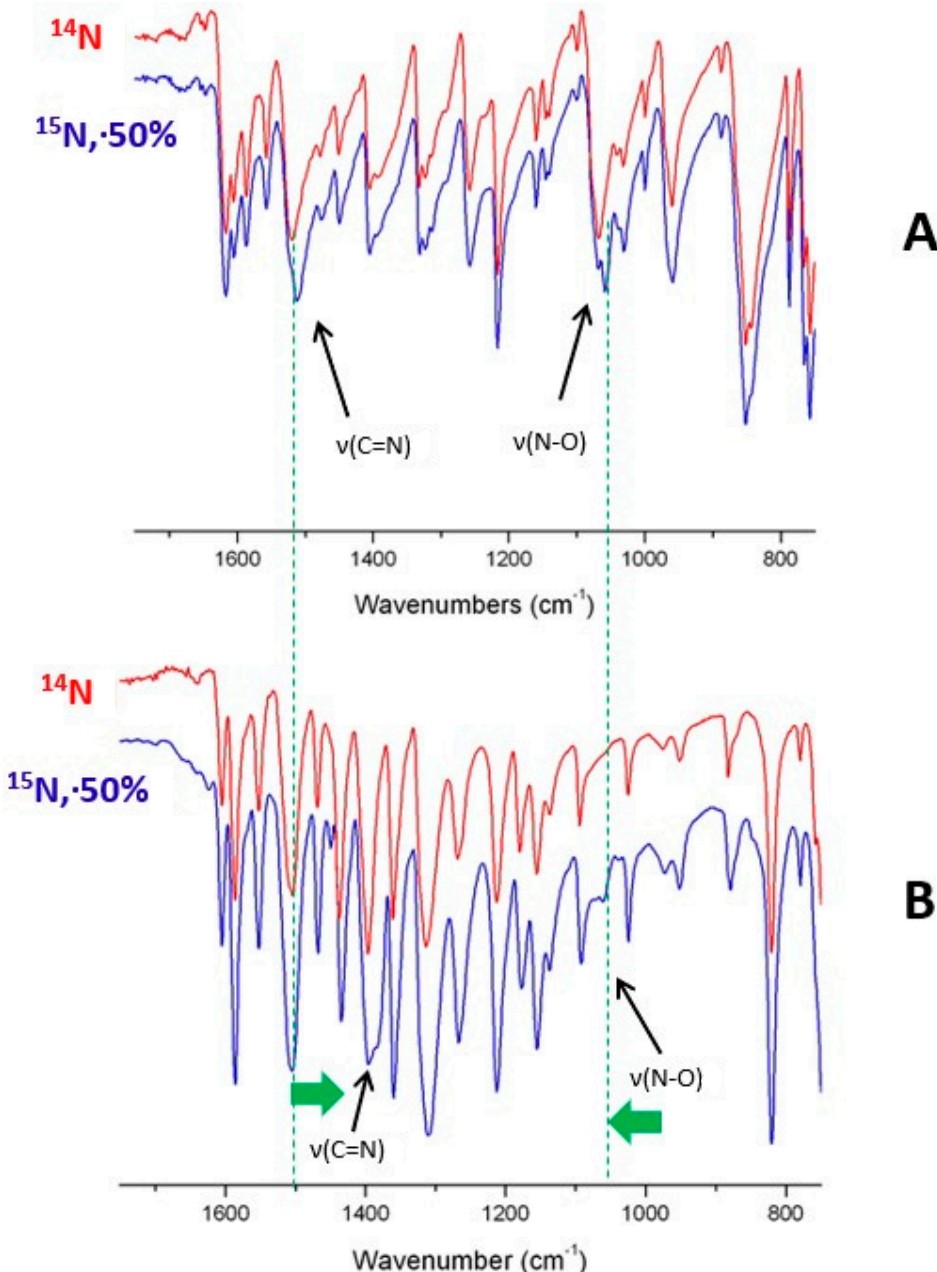

**Figure 3.** An overlay of the IR spectra of [15]N-labeled and unlabeled HL[1] ligand (**A**) and its Pt(II) complex PtL[1]$_2$ (**B**). Dotted lines show where peaks of the CNO fragment were in spectrum of uncomplexed ligand, while arrows indicate split bands due to the presence of isotope.

It should be noted that there is a significant shift in the $\nu$(C=N) and $\nu$(N-O) bands' positions from uncomplexed free ligands and Pd,Pt-complexes, due to the redistribution of electron density in this fragment. Thus, the pure *oxime* character of the >C=N-OH group changes to a *nitroso* character in coordinated anionic ligands, as shown in Scheme 2, and is a good agreement with crystal structures of the ligands and complexes we discussed above. The actual IR-spectra investigated in this work, ligands, and Pd, Pt complexes are presented in Supporting Information S29–S34. An adoption of the nitroso character of the group leads to the elongation of the C-N bond and shortening of the N-O bond in complexes, which is reflected in the decrease of the frequency of the $\nu$(C=N) vibration and slight increase in the $\nu$(NO) vibration. Very same behavior was observed earlier in the cyanoxime-based complexes of Tl(I), Ag(I), and Pd/Pt [50]. As can be seen from Table 5, the complexes of Pd and Pt with naphtoquinone oximes **L**[1] and **L**[2] have similar spectroscopic pattern, and

we postulate their similar coordination environment. Indeed, almost identical structures demonstrate monomeric complexes of these metals with cyanoximes [38–40].

### 3.3.2. Electronic Spectra

These were recorded as diffusion reflectance from powdery solid samples of Pd, and Pt naphtoquinoneoximes evidenced broad bands across the whole spectrum from 350 to 850 nm, which explains the very dark colors of compounds (Supporting Information S1,S2). These transitions belong to the CT-bands typical for complexes with conjugated chelating acidoligands. Based on data of electronic spectroscopy (Supporting Information S35), photoluminescence from these samples was measured, using several excitation wavelengths. The emission of both Pt-containing samples was observed at RT at ~985 nm using 750 and 800 nm excitation wavelengths, as displayed in Figure 4. No photoluminescence in the NIR region of spectra was detected from the powdery samples of the pure starting **HL$^1$** and **HL$^2$** ligands or from their solutions. Additionally, no photoluminescence from Pd-naphtoquinone oximes **PdL$^1_2$** and **PdL$^2_2$** in solid state or in solutions in organic solvents were observed, suggesting that the emission is entirely the property of complexes of platinum(II).

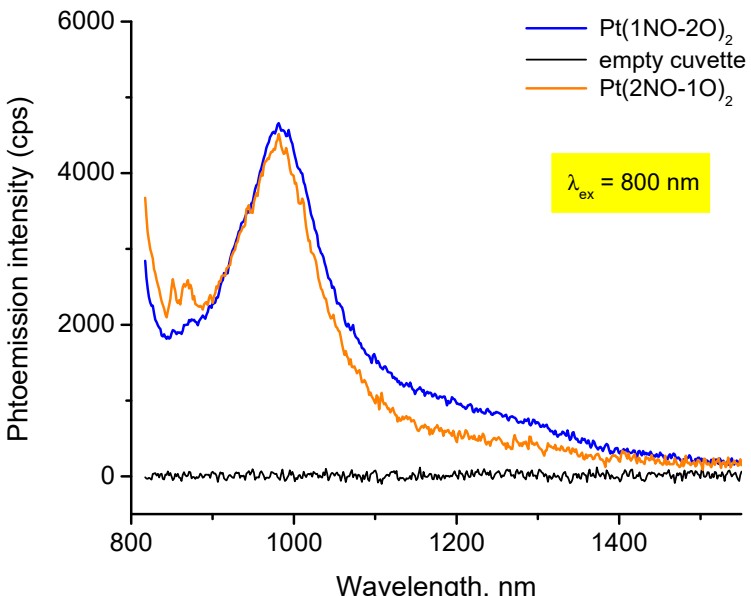

**Figure 4.** Photoluminescence spectra of solid powders of Pt(II)-naphtolate-oximes at 295 K drawn in unnormalized ('as is') fashion, in comparison with background of the instrument (cuvette, + black cardboard sample holder).

The exact origin of such low energy photoluminescence from 1D polymeric oximes-based complexes is not clear at this time, but we may suggest that it is a MLCT transition similar to several other Pt-containing complexes that were investigated in the past [51–55]. However, we observed in our current work that the emission is on 200–300 nm higher wavelengths, with significantly lower energy. The detected photoluminescence temporarily increased when the bulk powders of **PtL$^2_2$** and **PtL$^1_2$** were exposed to Br$_2$ vapors, which indicates a partial oxidation of Pt(II) centers to Pt(IV). This event facilitates the formation of a luminescent mixed valence 'metal wire' analogous to that previously found in Pt-cyanoximates [38,40]. Prolonged exposure of Pt-naphtoquinone-oximes to bromine leads to the degradation of the complexes, due to the facile ligands' halogenation.

We recently reported the detection of photoluminescence in the NIR region of spectra of solid molecular Pt-based oximates in a series of publications [38–40]. In those systems, organic ligands were mono-anionic deprotonated cyanoximes that also favored the adoption of the *cis*-geometry in the complexes formed. Monomeric complexes of PtL$_2$ com-

position (L = chelating cyanoximes MCO⁻ [32], PiPCO⁻ [55], PyrCO⁻ [39], 2PCO⁻ [19]) self-assembled in solutions with the formation of flexible 'molecular slinky' stacks [56] and formed in solid state dark green 1D 'poker chip' columns. Moreover, solid powders of these samples conduct electricity at the high end of semiconductors [38]. The latter fact evidences the formation of mixed valence Pt(II)/Pt(IV) wires that are quite similar in behavior to the famous tetracyano-platinates (KCP systems) investigated in the past [48–50,57–59]. However, those $[Pt(CN)_4]^{x-}$ 1D polymers do not emit in the NIR region of spectra. Therefore, the isomeric Pt-naphtoquinoneoximates prepared and characterized in this work represent an addition to this very rare and unusual family of solid state molecular NIR-emitters [51,60]. We cannot exclude the possibility of the formation of mixed valence Pt(II)/Pt(IV) pairs in our complexes. Indeed, in case of our complex **PtL²₂·CH₃CN**, the check CIF indicated presence of positive electron density of +1.89 $eA^{-3}$ on the Pt1 center that triggered B-type alert. Similarly, in the structure of the **PtL²₂·C₆H₅NO₂** complex, there is +2.02 $eA^{-3}$ electron density on Pt1 atom, which generated an A-type alert in the checkCIF report (Supporting Information S38). All that evidenced presence of partially oxidized metal centers originated most likely due to the fact that the air oxidation of Pt(II) indeed could happen and maybe facilitated by extensive π-back bonding from the conjugated aromatic ligand. However, further investigation of the oxidation states of Pt in the prepared complexes and their solid-state electrical conductivity was out of the scope of the current work. Additionally, we clearly detected the crucial effect of self-aggregation and formation of a 1D polymeric structure that directly affected the photoluminescence of these compounds. Quite similar behavior for several organic metal-free fluorophores, which exhibit photoemission induced by aggregation of molecules in stacks, albeit at much lower wavelengths, was recently observed [61,62]. Additionally, in the context of discussing the practicality of studied compounds, it is interesting to mention the development of fluorescent and colorimetric chemosensors for cations based on other aromatic naphthalene derivatives [62,63].

### 3.4. Hirshfeld Surface Analysis

As said above, two crystal structures contained stoichiometric molecules of solvents in the ASU in quite peculiar positions close to coordinated anionic oximes. In order to investigate the intermolecular interactions for both complexes, Hirshfeld surface analysis was performed using CrystalExplorer software (version 17.5) [56,63]. The calculation was based on the CIF files from the X-ray crystal structures of those two complexes.

Interesting π–π stacking interactions were observed among the neighboring units of complex **PtL²₂·CH₃CN**. As shown in Figure 5, the blue dashed lines represent the nearly perfect face-to-face overlap of the aromatic rings, with a centroid–centroid distance of 3.365 and 3.373 Å, and the green lines represent the weaker interactions among the offset aromatic rings with a much longer centroid–centroid distance of 3.728–3.761 Å. These π–π stacking interactions can be verified by the Hirshfeld surface mapped with surface index (Figure 6A left). Both sides of the complex **PtL²₂·CH₃CN** show an identical pattern, with red/blue triangle pairs, which is a characteristic feature of π–π stacking. The surface area contributed by these π–π stacking interactions is 16.8% of the total surface area of complex **PtL²₂·CH₃CN**, which shows in the center of the two-dimensional fingerprint plot (Supporting Information S36). Interestingly, the Hirshfeld surface mapped with $d_{norm}$ for both sides of the planar molecule shows a weak red dot on the platinum center (Figure 6A right), suggesting a weak interaction between Pt centers. In fact, the X-ray crystal structure of complex **PtL²₂·CH₃CN** shows a Pt–Pt distance of 3.381 and 3.497 Å, which sits in the range of effective metal–metal interactions (3.09–3.50 Å) [57,58,64].

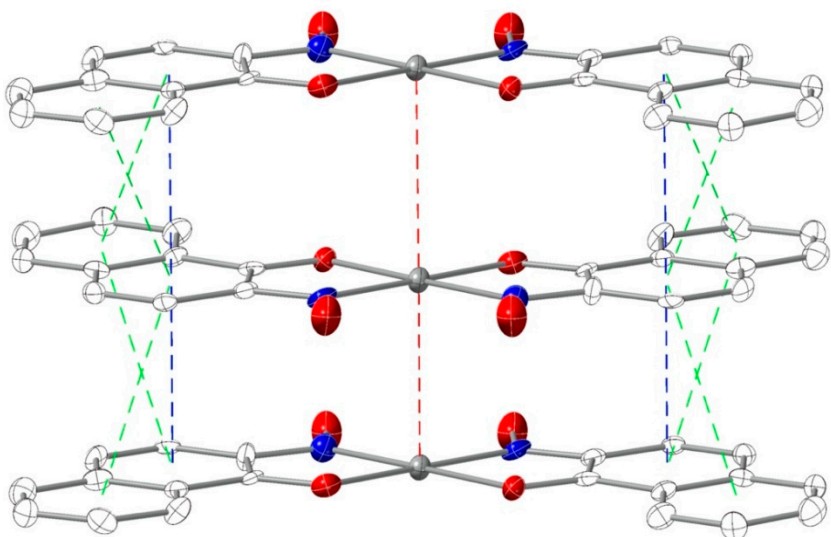

**Figure 5.** Intermolecular $\pi$–$\pi$ stacking interactions and Pt–Pt interactions in the pack structure of complex 'Two-domains-only' (complex **PtL$^2$$_2$·CH$_3$CN**). Coloring scheme: red—O, blue—N, grey—Pt, white—C atoms.

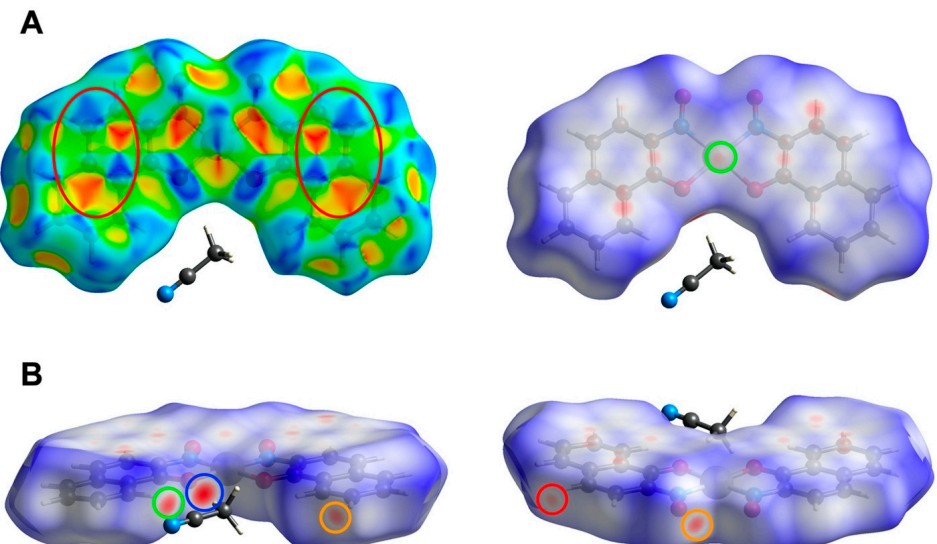

**Figure 6.** (**A**)—Hirshfeld surface of complex **PtL$^2$$_2$·CH$_3$CN** mapped with surface index (**left**) and $d_{norm}$ (**right**) showing the $\pi$–$\pi$ stacking interactions and Pt–Pt interactions. (**B**)—Hirshfeld surface of complex **PtL$^2$$_2$·CH$_3$CN** mapped with surface index (**left**) and $d_{norm}$ (**right**) showing the hydrogen bonding interactions.

In addition to the $\pi$–$\pi$ stacking interactions, interesting intermolecular interactions involving hydrogens are also observed among complex **PtL$^2$$_2$·CH$_3$CN** units and acetonitrile molecules (Figure 7). One of the hydrogens from an acetonitrile molecule interacts with the C–O oxygen closely (blue lines, $d_{C-H\cdots O}$ = 2.386 Å; $\angle_{C-H\cdots O}$ = 165.6°). On the other side, the ligand backbone, the oxime oxygen interacts with one of the C–H groups from the neighboring unit (orange lines, $d_{C-H\cdots O}$ = 2.529 Å; $\angle_{C-H\cdots O}$ = 150.1°). A weak interaction with a larger distance is also observed between a C–H group and acetonitrile nitrogen atom (red lines, $d_{C-H\cdots O}$ = 2.628 Å; $\angle_{C-H\cdots O}$ = 145.8°). Meanwhile, a weak T-shaped C–H$\cdots\pi$ interaction between the aromatic C–H group and the C$\equiv$N carbon is observed (green lines) [65,66]. These interactions can be confirmed on the Hirshfeld surface mapped with $d_{norm}$ (Figure 6B). The red dots highlighted by the colored circles correspond to the colored lines in Figure 6.

The molecules of complex **PtL$^2_2$·C$_6$H$_5$NO$_2$** exhibit a slightly different packing structure from complex **PtL$^2_2$·CH$_3$CN**. Among the three units shown in Figure 8, the middle unit interacts with the top unit through multiple $\pi$–$\pi$ stacking interactions, with either face-to-face overlap (blue lines, centroid–centroid distance = 3.342 Å) or offset fashion (green lines, centroid–centroid distance = 3.701 and 3.782 Å). Meanwhile, the middle unit interacts with the bottom unit through the offset $\pi$–$\pi$ stacking interactions only (green lines, centroid–centroid distance = 3.467 Å). This difference suggests a stronger contact of the top pair, in which the two units are almost aligned along the crystallographic $x$ axis, leading to a Pt–Pt interaction with a metal–metal distance of 3.307 (red lines) that is shorter than both Pt–Pt distances in complex **PtL$^2_2$·CH$_3$CN**. However, for the bottom pair, the displaced parallel conformation gives a much longer Pt–Pt distance of 3.553 Å, which is above the upper limit of Pt–Pt interaction (3.50 Å) reported in the literature [59,60,65,66].

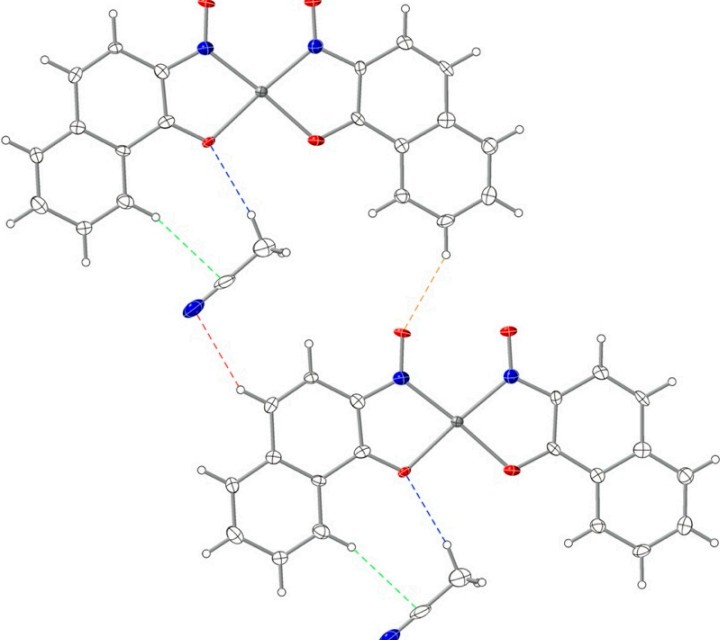

**Figure 7.** Intermolecular interactions involving hydrogens in complex **PtL$^2_2$·CH$_3$CN**. Coloring scheme: red—O, blue—N, grey—Pt, white—C atoms.

The observation of intermolecular interactions in the crystal packing of complex **PtL$^2_2$·C$_6$H$_5$NO$_2$** is well-supported by the Hirshfeld surface analysis. Unlike complex **PtL$^2_2$·CH$_3$CN**, the Hirshfeld surface of the middle unit in Figure 8 shows a very different pattern on two sides of the planar molecule. On the top side (Figure 9A left), the multiple characteristic red/blue triangle pairs suggest the presence of multiple $\pi$–$\pi$ stacking interactions between the top two units, while on the bottom side of the middle unit (Figure 9A right), only two pairs of red/blue triangles are observed, indicating the presence of less $\pi$–$\pi$ stacking interactions. This matches with the stacking pattern observed in the X-ray crystal structure of complex **PtL$^2_2$·C$_6$H$_5$NO$_2$** (Figure 8). Based on the fingerprint plot shown in Supporting Information S37, the $\pi$–$\pi$ stacking interaction contributes 13.7% of total surface area, less than the 16.8% of complex **PtL$^2_2$·CH$_3$CN**, suggesting less C–C contact, due to the larger distance and displaced conformation between the bottom two units. On the Hirshfeld surface mapped with $d_{norm}$ for the middle unit, there is a red dot right above the Pt center on the top of the planar molecule (Figure 9B left, green circle), suggesting the presence of a Pt–Pt interaction between the middle and top units, while on the other side of the middle unit, no red dots on the Pt center are observed (Figure 9B, right).

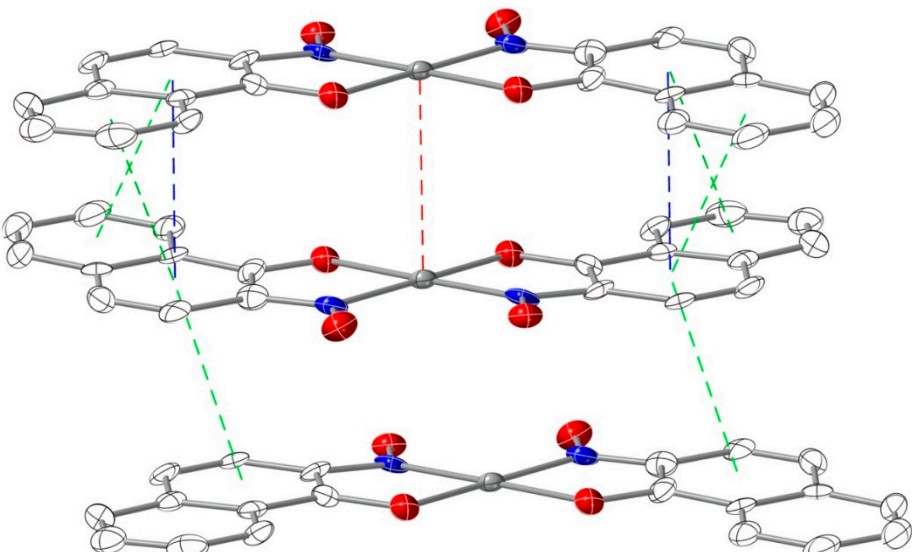

**Figure 8.** Intermolecular $\pi$–$\pi$ stacking interactions and Pt–Pt interaction in the pack structure of complex **PtL$^2_2$·C$_6$H$_5$NO$_2$**.

In addition to the $\pi$–$\pi$ stacking interactions, the oxygen atom of one oxime group in complex **PtL$^2_2$·C$_6$H$_5$NO$_2$** interacts with the nitrobenzene molecules through hydrogen bonding interactions (sch 10 left). On the Hirshfeld surface mapped with $d_{norm}$ (Figure 10 right), there are two red spots highlighted by green circles on the oxime oxygen atom, representing the two hydrogen bonds on the X-ray crystal structure (blue lines, $d_{C-H\cdots O}$ = 2.659 Å, $\angle_{C-H\cdots O}$ = 108.2°; $d_{C-H\cdots O}$ = 2.500 Å; $\angle_{C-H\cdots O}$ = 132.0°). The angles are smaller than conventional C–H$\cdots$O interactions (mean value = 152°), and only a few examples with such weak directionality of C–H$\cdots$O interaction are reported [61,67].

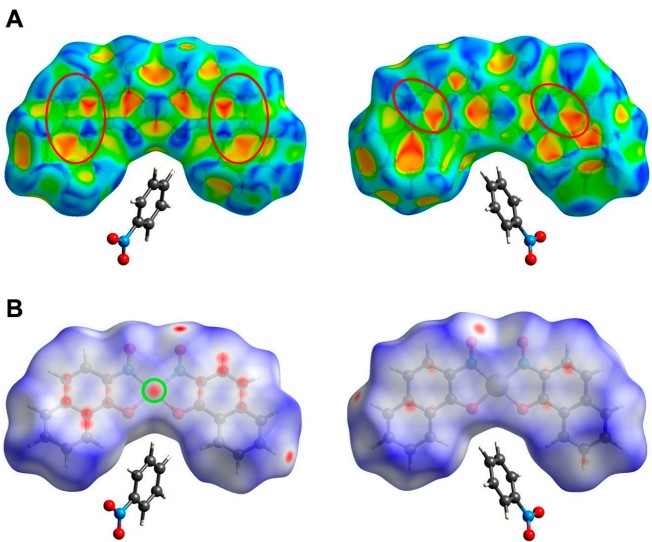

**Figure 9.** (**A**)—Hirshfeld surface of the middle unit mapped with surface index (**left**: top side; **right**: bottom side) showing the $\pi$–$\pi$ stacking interactions. (**B**)—Hirshfeld surface of complex **PtL$^2_2$·C$_6$H$_5$NO$_2$** mapped with $d_{norm}$ showing the Pt–Pt interaction (**left**).

Figure 11 shows another set of hydrogen bonding interactions among two nitrobenzene molecules and one Pt complex unit. The blue lines represent the strong interaction between a pair of nitrobenzene molecules that are coplanar to each other ($d_{C-H\cdots O}$ = 2.461 Å; $\angle_{C-H\cdots O}$ = 150.4°), which correspond to the two red dots highlighted by blue circles on the Hirshfeld surface of a nitrobenzene molecule mapped with $d_{norm}$. The third C–H$\cdots$O

interaction (green line) occurs between the oxygen atom from one nitrobenzene molecule and one C–H group from ligand backbone ($d_{\text{C–H}\cdots\text{O}}$ = 2.563 Å; $\angle_{\text{C–H}\cdots\text{O}}$ = 148.7°), and it can be identified as the red dot highlighted by the green circle in Figure 11 (left).

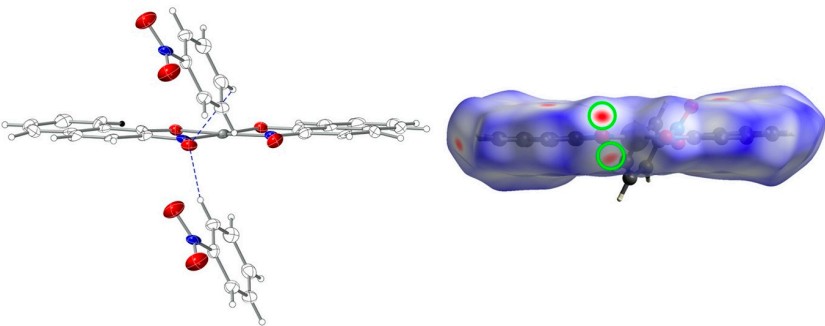

**Figure 10.** **Left**: weak hydrogen bonding interaction involving the oxime group of complex **PtL$^2{}_2$·C$_6$H$_5$NO$_2$** and nitrobenzene molecules. **Right**: Hirshfeld surface of complex **PtL$^2{}_2$·C$_6$H$_5$NO$_2$** mapped with $d_{\text{norm}}$.

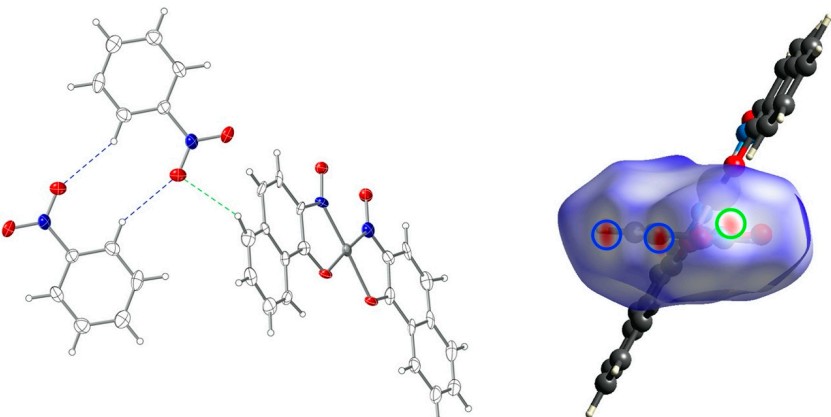

**Figure 11. Left**: weak hydrogen bonding involving nitrobenzene molecules and complex **PtL$^2{}_2$·C$_6$H$_5$NO$_2$**. **Right**: Hirshfeld surface of nitrobenzene mapped with $d_{\text{norm}}$.

## 4. Conclusions

As a summary of the investigation carried out, we can state the following:

Four coordination compounds based on planar aromatic conjugated isomeric naphtoquinone oximes and Pd(II), Pt(II) were synthesized and characterized for the first time using a variety of physical and spectroscopic methods.

The thermal stability of the starting oximes and their metal complexes was examined, and the data indicated their unusually strong exothermic decomposition.

Two crystal structures of Pt-complexes with 2-oxime-1-quinone containing stoichiometric solvents molecules (acetonitrile and nitrobenzene) were determined and showed an unusual orientation of solvents towards oxygen atoms of the *phenol* and/or *nitroso*-group of coordinated anion because of non-covalent C-H—O electrostatic interactions.

In both crystal structures, the Pt center adopts the *cis* geometry of coordinated naphtoquinone oximes and these Pt-complexes represent a 1D columnar solid with pronounced metallophilic interactions that form 'metal wire'.

Photoemission in the NIR region of the spectrum from solid Pt-complexes with both isomeric planar aromatic naphto-quinone oximes was determined, but not for similar Pd-complexes.

The Hirshfeld surface analysis was performed for both of the studied complexes and showed the presence of an interesting intermolecular interaction network. The formation

of the $\pi$–$\pi$ stacking interactions and Pt–Pt interactions among the mononuclear units could be attributed to the planar conformation of the coordination molecules.

**Supplementary Materials:** The following supporting information can be downloaded at: https://www.mdpi.com/article/10.3390/inorganics11030116/s1, Actual photographs of powdery samples of Pt(II) and Pt(II) naphtoquinone-oximes under x40 magnification (S1,S2); elemental analyses data for synthesized compounds (S3–S5); crystals used in this study (S6); Twin laws (transformation matrices for the first 4 domains) for crystal sample of **PtL$_2$$^2$**, acetonitrile solvate (S7); Twin laws (transformation matrices) for crystal sample of **PtL$_2$$^2$**, nitrobenzene solvate (S8); residual electron densities maps showing Q-peaks around Pt-centers (S9,S10); thermogramms for pure organic ligands (S11–S14); traces of thermal analyses for metal complexes (S15–S18); metallic Pd and Pt sponges as results of thermal decomposition of synthesized complexes (S19); geometry of square-planar environment of Pt center in **PtL$_2$$^2$**, CH$_3$CN-solvate (S20); prospective view along a direction of the unit cell content in the structure of **PtL$_2$$^2$**, acetonitrile solvate showing: the formation of the "head-to-tail" $\pi$–$\pi$ stacked dimer and solvent molecules in the unit cell indicated by arrows (S21); pruned view of several unit cells along c direction showing the geometry of "Pt-wires" in the structure of of **PtL$_2$$^2$**, acetonitrile solvate (S22); the ASU in the structure of **PtL$_2$$^2$**, acetonitrile solvate showing closest electrostatic contacts between solvent and the anion in the complex (S23); geometry of square-planar environment of Pt center in **PtL$_2$$^2$**, C$_6$H$_5$NO$_2$-solvate (S24); prospective view along a direction of the unit cell content in the structure of **PtL$_2$$^2$**, nitrobenzene solvate showing the formation of the $\pi$–$\pi$ stacked "head-to-tail" dimer and solvent molecules electrostatically connected to the anions in complex (S25); pruned view of several unit cells along c direction showing the geometry of "Pt-wires" in the structure of **PtL$_2$$^2$**, nitrobenzene solvate (S26); details of geometry of intercalated solvent molecule in the crystal structure (S27,S28); IR-spectra of pure organic isomeric naphtoquinone oxime ligands and their Pd, Pt complexes (S29–S34); electronic spectra (DRS) of Pt-complexes (S35); two-dimensional fingerprint plot of complex **PtL$_2$$^2$·CH$_3$CN** showing the contribution of $\pi$–$\pi$ stacking interaction to the total surface area (S36); two-dimensional fingerprint plot of complex **PtL$_2$$^2$·C$_6$H$_5$NO$_2$** showing the contribution of $\pi$–$\pi$ stacking interaction to the total surface area (S37); checkCIF reports for crystal structures (S38).

**Author Contributions:** Conceptualization, N.G. and S.T.; methodology, N.G.; Hirshfield surface analysis software use, L.Y.; formal analysis, L.Y.; investigation, M.M. and C.T.; resources, M.B.; data curation, N.G. and L.Y.; writing—original draft preparation, N.G. and L.Y.; review and editing, M.B.; supervision, N.G.; project administration, N.G. All authors have read and agreed to the published version of the manuscript.

**Funding:** This research received no external funding.

**Data Availability Statement:** Crystal and refinement data for two reported in this paper new structures can be retrieved from CCDC (England) with inquiries under numbers 2,240,578 (acetonitrile solvate) and 2,240,577 (nitrobenzene solvate). Actual spectra of studied compounds and some quantities (not exceeding 200 mg) of reported Pd and Pt complexes are available from corresponding author upon request.

**Acknowledgments:** NG is very grateful for help of Emma Fonke in recording the solid-state UV–visible spectra (diffuse reflectance from powders) of all the complexes presented here, to the MSU College of Natural and Applied Sciences for support of the X-ray diffraction laboratory, and to Victoria Barry for invaluable technical assistance. Additionally, NG thanks Michael Hilton for his initial work during the inception of this project and for making $^{15}$N labeled ligands with subsequent recording of their NMR and vibrational spectra as part of the complete characterization of these remarkable ligands.

**Conflicts of Interest:** The authors declare no conflict of interest in the submitted work.

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
