# Peer review of "Synthesis and Characterization of Pt(II) and Pd(II) Complexes with Planar Aromatic Oximes"

_inorganics, doi:10.3390/inorganics11030116_

Round 1

Reviewer 1 Report

The manuscript reports an investigation of four new metal-organic compounds based on quite rare keto-oximato ligands. The text is entirely well wirtten and discussions are clear. The overall study looks very complex and comprehensively worked out. As a considerable disadvantage of the work, it may be noticed that only for one of four complexes its molecular structure was clearly determined in fact (within two different solvates). However, I highly appreciate an obviously very strong author's effort to the structure solution and refinement on the basis of very complicated initial data taken from the poor-quality crystals. Major crititcal notes are the following: 

1. As mentioned above, no real structure confirmation exists for three of four characterized complexes. TG-DSC results for Pd-L2 complex might evidence its individuality, but I guess this is not enough to make fully reliable conclusions. Authors could perform powder XRD for Pd-L2 to confirm its purity and compare its structure to Pt-L2, at least qualitatively. Concerning complexes based on L1, I would recommend to delete them from the paper, taking into account that no any real (even qualitative) structural information has been obtained for them. Or their additional characterization (by means of PXRD, NMR or any other valuable method) might give an important structural information. 

2. Strings from the cif file of Pt-L2 nitrobenzene solvate: 

_diffrn_reflns_theta_max 23.439
_diffrn_reflns_theta_full 25.242

look very strange, as well as their derived values of completeness. I can suggest that the shown numbers are confused to each other and crystallographic tables are built from the incomplete cif files. Please check and revise the data if necessary. 

Author Response

...please see attached file. Thank you for diligent and thorough reading! 

Reviewer 2 Report

The paper reports a family of square-planar palladium(II) and platinum(II) complexes with oxime derivative ligands. The NIR photoluminescence behavior of the solid Pt complexes and thermal analysis results, as well as detailed crystal structural analysis data, were described in the manuscript. 

I am very worried about the discussion about the cis-trans isomerism in this manuscript. The single-crystal X-ray analysis represent the cis isomer of the complexes, but the paper did not confirm the selective formation of the cis-isomer by spectroscopic techniques. I think 1H NMR spectral measurement is nessesary to decide the bulk sample of complexes contains only one isomer or a mixture of cis and trans isomer.

No major revision is needed if the NMR spectral data supports the selective formation of the cis isomer for the bulk sample. However, if the NMR indicated the presence of two isomers in the bulk samples, the authors have to re-measure the photoluminescence spectra using crystalline sample which contains only the cis isomer. 

Author Response

please see reply to Reviewer 2 comments/critique

Reviewer 3 Report

The manuscript submitted by the authors reported the synthesis of a series of metal (Pd and Pt) naphtoquinone oxime complexes.  The products were characterized and the results are convincing. I recommend publication after minor revision. 

1.     I would suggest the authors to summarized the characterization data for each complex in the experimental section。

2.     The nitroso group in the complexes should show a medium to strong characteristic band in IR.  Any findings?

Author Response

Thank you for reading of our manuscript!

yes, this time in R1 version we included a substantial part of the vibrational spectra discussion. 

we believe that manuscript should be structured in a classic way: experimental part, then results-&-discussion, then conclusions, and then citations. A combination of compounds characterization + preparation (experimental part) would create undesired clutter and confusion to the reader [at least in our opinion].

Reviewer 4 Report

In this manuscript the authors stated to report the preparation and characterization of four platinum(II) and palladium(II) complexes bearing naphtoquinone oximes. However, this reviewer found that the most part of the manuscript described crystal structures (and the Hirshfeld Surface Analysis) of one of the Pt complex (with two different solvate molecules). For the other complexes, Pd(L1)2, Pd(L2)2 and Pt(L1)2, more detailed descriptions are required for full characterization of the compounds. For example, the elemental analysis for the complexes were not reported (even in SI). Only limited spectral data were described in SI; at least, the IR spectra of all compounds should be reported. The powder X-ray diffraction could give some information for crystal structures of the complexes. In my opinion, the authors must revise this manuscript thoroughly based on the science they want to mention to the society. 

In addition, the following points should be noticed in the revision. 

(1)  The solvated molecule interacts to the oxygen atoms of (not the nitroso group, but) the ketone. 

(2)  Chart 1 should be modified. At least, acetonitriles is not appropriate. 

(3)  The description of crystallographic work (pages 4 to 6) is too much detailed. Some of them should be moved to SI. 

(4)  Table 1 is not completed. Check carefully. Also, in Table 3 N2-O3 are duplicated. 

(5)  What would happen in the first decomposition step of thermal analysis?

(6)  The conclusion are bulleted, and must be modified as clear as possible. 

(7)  The styles of referenced journals are not uniformed. 

Author Response

please see file attached. thank you for reading of our manuscript!

Round 2

Reviewer 1 Report

Authors did their best in the revision of the manuscript. Appended by new discussions on the basis of additional experimental data and author's valuable comments, the paper now looks fully acceptable for publication and I can recommend it with great pleasure. 

Author Response

Thank you for careful reading and reviewing our manuscript!

Reviewer 2 Report

The reviewer understand the character of compounds which is poorly soluble for the NMR measurements.

The authors addressed all the other comments from reviewers. So, I recommend the paper for publication in the present form.

Author Response

Many thanks for reading of our manuscript!

Reviewer 4 Report

The revised paper became better shape, and the authors answered my previous concerns precisely. The followings are minor questions and comments, which authors should address further in the  further but minor revision.

(1) If the authors want to claim the electrostatic C–H···O contacts with both nitroso- and naphthyl groups in the crystal structure of Pt(L1)2·C6H5NO2, Figure S25 A (or the equivalent) is much suitable than Figure 2 A. 

(2) “Substituted acetonitriles” or “Nitriles” would be suitable in this case. 

(4) Check the numbers in Table 1 for Volume, the index range, and Reflections collected for the CH3CN adduct. 

(6) The conclusion should (not be bulleted but) be summarized in one or a few paragraphs.

(7) Check the journal abbreviations for Refs. 12, 24, 26, 27, 29, 37–39, 64, 65, 67, 68.

(a) I still wonder why only partial data are provided in this paper (or the attached supplementary material); for example, missing the EA data for Pd(L2)2, no TA data for both Pt and Pd complexes of L1. 

(b) In Figure 3, the positions of arrows should be wrong. 

Author Response

Thank you for careful and thorough reading of our manuscript!
